# Improving GANs with A Dynamic Discriminator

**Ceyuan Yang**[1,3,†]     **Yujun Shen**[2,†]     **Yinghao Xu**[1]     **Deli Zhao**[2]     **Bo Dai**[3]     **Bolei Zhou**[4]

[1]CUHK     [2]Ant Group     [3]Shanghai AI Laboratory     [4]UCLA

## Abstract

Discriminator plays a vital role in training generative adversarial networks (GANs) via distinguishing real and synthesized samples. While the real data distribution remains the same, the synthesis distribution keeps varying because of the evolving generator, and thus effects a corresponding change to the bi-classification task for the discriminator. We argue that a discriminator with an *on-the-fly* adjustment on its capacity can better accommodate such a *time-varying* task. A comprehensive empirical study confirms that the proposed training strategy, termed as *DynamicD*, improves the synthesis performance without incurring any additional computation cost or training objectives. Two capacity adjusting schemes are developed for training GANs under different data regimes: i) given a sufficient amount of training data, the discriminator benefits from a progressively increased learning capacity, and ii) when the training data is limited, gradually decreasing the layer width mitigates the over-fitting issue of the discriminator. Experiments on both 2D and 3D-aware image synthesis tasks conducted on a range of datasets substantiate the generalizability of our DynamicD as well as its substantial improvement over the baselines. Furthermore, DynamicD is synergistic to other discriminator-improving approaches (including data augmentation, regularizers, and pre-training), and brings continuous performance gain when combined for learning GANs.[1]

## 1 Introduction

Generative adversarial network (GAN) [16], which consists of a generator and a discriminator, significantly advances image generation. In general, these two components compete against each other during training. The generator aims to emulate the observed data distribution through producing as realistic images as possible, and the discriminator learns to differentiate fake samples from real ones and guides the generator towards better synthesis. Despite the great effort of improving GANs from the generator side [40, 60, 29, 31, 32, 5], it is relatively less explored on the important role of the discriminator in this two-player game. In fact, discriminator is the one that accesses the training data, examines how close the real and synthesis distributions are, and derives loss functions to train both itself and the generator. Therefore, learning an apt discriminator is also essential for GANs.

The discriminator in a GAN is typically learned with a bi-classification task. It aims to categorize images into two folds depending on whether they come from the training set or are synthesized by the generator. Existing studies on image classification [19, 20] have pointed out, it is critical to align the model capacity to the task difficulty, otherwise the issue of either under-fitting or over-fitting occurs. For instance, ResNet-50 [19] performs worse than ResNet-101 on ImageNet classification [11] because it is not capable enough to handle the data variations. Nevertheless, ResNet-152 outperforms ResNet-200 on the same task, where the latter model has too many parameters and thus over-fits the training set [20]. From this perspective, the capacity of a GAN discriminator as the classifier should be also aligned with the aforementioned bi-classification task.

---

† denotes equal contribution.

[1]Code and models are available at https://genforce.github.io/dynamicd.

36th Conference on Neural Information Processing Systems (NeurIPS 2022).

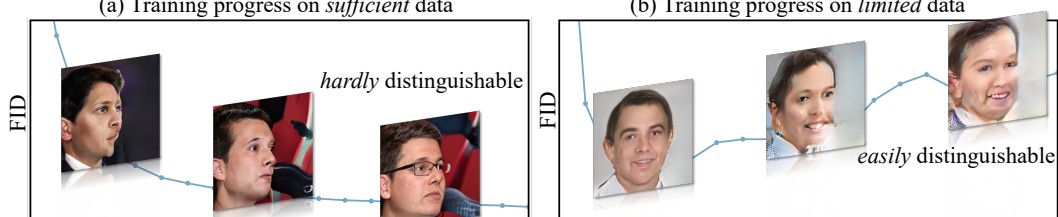

(a) Training progress on *sufficient* data  (b) Training progress on *limited* data

*hardly* distinguishable

*easily* distinguishable

Figure 1: **Illustration of the *time-varying* bi-classification task for the discriminator** under the training settings of (a) sufficient data and (b) limited data. Though the real data distribution is fixed, the synthesis distribution keeps varying during training due to the evolving generator. Samples with the same latent code produced from the generator at different training stages show the synthesis distribution shift. FID under different training epoch, which measures the similarity between real and fake distributions, indicates the varying difficulty of the bi-classification task.

Different from the common image classification tasks where the training data remains fixed during the whole training process, GAN training appears to be *time-varying* since the synthesis quality of the generator is constantly evolving, as suggested in Fig. 1. That way, although the real data distribution keeps the same, the varying synthesis distribution still results in the change of the bi-classification task for the discriminator. It naturally raises a question: *does a discriminator with a fixed capacity meet the demand of such a dynamic training environment?*

To answer this question, we conduct a comprehensive empirical study by training GANs with a *dynamic discriminator (DynamicD)*, where an *on-the-fly* adjustment is enforced on its model capacity during training. We first investigate a plain form where the layer width of the discriminator is linearly adjusted. Under such a setting, the generator supervised by our DynamicD achieves far better synthesis performance than its counterpart learned with a fixed discriminator, which is with either the starting capacity or the ending capacity.[2] It is noteworthy that our proposed training strategy is highly efficient as it relies on neither additional computing cost nor extra loss functions. Inspired by this, we come up with two capacity adjusting schemes and confirm that different training data regimes have different favored schemes. On one hand, with a sufficient amount of training data in Fig. 1a, the discrimination task becomes increasingly challenging when the generator gets more capable. In this case, the discriminator benefits from a enlarged capacity to match the generator. On the other hand, with limited training data in Fig. 1b, the longer the model is being trained, the closer the discriminator is to memorizing the entire dataset [30]. As a result, a scheme to gradually decrease the model capacity assists the discriminator against over-fitting.

We evaluate our method on both tasks of 2D image synthesis and 3D-aware image synthesis. On a wide range of datasets including human faces [29], animal faces [10], scenes [58], and synthetic cars [12], DynamicD exhibits consistent improvements over the baselines. Furthermore, we show that DynamicD is synergistic to existing approaches that improve GAN discriminator, including data augmentation [30], training regularizers [61], and pre-training [41]. It brings extra performance gain when combined and opens a new dimension in improving GAN training.

## 2   Related Work

**Generative adversarial networks.** Recent efforts on architectural improvements [45, 28, 5, 29, 31, 32, 36, 2, 26, 13] and training methods [3, 18, 40, 39] provide the appealing synthesis result, even 3D controllability [48, 43, 8, 17, 56]. Based on these, various techniques are proposed to manipulate semantics [15, 49] and edit real images [1, 65, 46]. In addition, GANs can also improve various discriminative tasks in turn [23, 55, 7, 44]. In this work, we aim at exploring the dynamic capacity of discriminator at one fundamental view. Some related work is the progressive growing training [28, 37] which adjust the generator and discriminator accordingly from low-resolution to high-resolution. Differently, we do not modify the generator and only focus on studying the capacity of the discriminator.

---

[2]Experimental setup and detailed analysis can be found in Sec. 4.2 and Tab. 1.

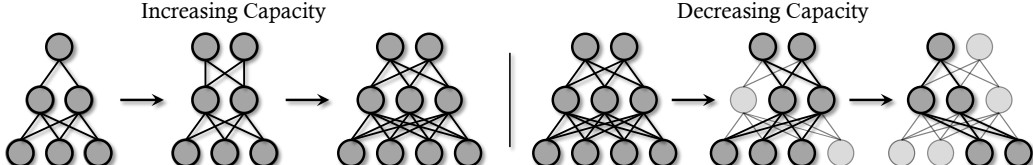

Figure 2: **Two schemes for *on-the-fly* capacity adjustment in DynamicD.** *Left:* We gradually increase the network width via including newly initialized filters. *Right:* We progressively decrease the network width by *randomly* dropping a subset of filters. "Random" means that, even under the same capacity, the discriminator may use different filers at different training steps.

**Improving discriminator in GANs.** Many attempts have been made in improving discriminator from various perspectives. Some literature [64, 53, 62, 30, 25] explore how data augmentation can help alleviate the ovefitting of discriminator, which works perfectly under low-data regime. However, the improvement becomes limited even negative given sufficient training data. Meanwhile, prior work also take efforts to either incorporate kinds of regularization [61, 63, 39] or introduce various extra tasks [9, 52, 24, 27, 59, 57, 54] for discriminator. Although a discriminator could be indeed enhanced to some extent, extra computations are unavoidable. Recently, researchers start to make the best of the pre-trained models on large-scale data collection (*e.g.,* ImageNet [11]) as a frozen feature extractor of discrimintor. Sauer *et al.* [47] proposed that pre-trained feature space with projection could significantly improve convergence speed. Meanwhile, Kumari *et al.* [34] improved GAN training by ensembling multiple off-the-shelf models. Nevertheless, the most recent work [35] suggests that using ImageNet pre-trained models might make the metrics unreliable in practice. Different from prior work, we focus on adjusting capacity of discriminator on-the-fly, to align with the time varying bi-classification task. Such that, synthesis under different data regimes could be further improved without extra computation cost. We also show that the proposed method is synergistic to these existing discriminator-improving techniques and brings consistent performance gain when combined.

**Model augmentation.** Different from data augmentation methods which directly operate on data, model augmentation methods augment neural representation directly. One representative example is Dropout [50] which randomly eliminates the units of a neural network to alleviate the over-fitting issue. A variety of dropout operations are proposed for better regularizations and performances, like SpatialDropout [51], DropBlock [14] and StochasticDepth [22]. Recently, Cai *et al.* [6] introduced network augmentation into training to improve tiny neural networks. Meanwhile, Liu *et al.* [38] demonstrated that model augmentation could work well with contrastive learning. The literature of model augmentation mostly focuses on improving discriminative models. Mordido *et al.* [42] proposed to involve multiple discriminators and then selected a subset of discriminators to train the generator. Differently, our approach focuses on one discriminator and investigates the effect of varying capacity from both decreasing and increasing perspectives.

## 3 Methodology

In the two-player competition of GANs, a discriminator aims at distinguishing real and synthesized images to accomplish bi-classification task. However, the synthesized data distribution varies with the evolving generator, thus the bi-classification task has a significant distribution shift issue. To tackle this, we propose to adjust the capacity of discriminator on-the-fly (called DynamicD) to match such a dynamically varying bi-classification task. With such a dynamic discriminator, the image synthesis quality under different data regimes could be further improved. In Sec. 3.1, we will briefly introduce the background of GAN training. Sec. 3.2 presents two schemes to dynamically adjust the capacity of discriminator, followed by a practical implementation under different data regimes in Sec. 3.3.

### 3.1 Preliminary

Generative Adversarial Network (GAN) [16] regards image synthesis as a two-player competition between a generator and a discriminator. Given a collection of observed data $\{\mathbf{x}_i\}_{i=1}^{K}$ with $K$ samples, the generator $G(\cdot)$ learns to map a randomly sampled latent code $\mathbf{z}$ which is usually subject to a pre-defined distribution $\mathcal{Z}$ (*e.g.,* normal distribution) to a realistic image. Meanwhile, a discriminator $D(\cdot)$ aims at distinguishing the observed image $\mathbf{x}$ sampled from observed data distribution $\mathcal{X}$ from the

synthesized $G(\mathbf{z})$ as a bi-classification task. These two models are optimized jointly in an adversarial manner:

$$\mathcal{L}_G = -\mathbb{E}_{\mathbf{z} \in \mathcal{Z}}[\log(D(G(\mathbf{z})))], \tag{1}$$

$$\mathcal{L}_D = -\mathbb{E}_{\mathbf{x} \in \mathcal{X}}[\log(D(\mathbf{x}))] - \mathbb{E}_{\mathbf{z} \in \mathcal{Z}}[\log(1 - D(G(\mathbf{z})))]. \tag{2}$$

Eventually, the generator could synthesize realistic images enough to confuse the discriminator. Since discriminator is the only one that could see the observed data, and measure how similar the observed and synthesized distributions are, it is essential to investigate the effect of capacity on the GAN training.

### 3.2 Dynamic discriminator

During the two-player competition, the synthesized data distribution keeps varying due to the evolving generator. It also makes the bi-classification task change accordingly. Therefore, the capacity of discriminator required by the varying bi-classification task might be also different as training goes by. Different from previous work that always uses a discriminator with fixed capacity, we propose to adjust the capacity of the discriminator dynamically, termed as DynamicD. Meanwhile, considering the synthesis under different data regimes might needs different dynamic capacities of discriminator, we propose two adjustment schemes for *increasing* and *decreasing* capacity respectively.

**Increasing capacity.** If the bi-classification task becomes challenging while we have a weak discriminator, under-fitting would occur, such that a generator with the relatively low synthesis quality could easily fool the discriminator. We thus progressively increase the capacity of discriminator by including newly initialized neural filters every several iterations. That is, assuming one layer $W_N^M$ containing $M$ neural filters with dimension $N$, increasing strategy aims at introducing $\alpha M$ extra filters where $\alpha$ denotes an extending coefficient.

Taking a convolution layer with a kernel of $M \times N \times 3 \times 3$ as an example, we would leverage another $\alpha M$ kernels with spatial size $3 \times 3$. Such that, combining the original kernel with the newly introduced ones, we could easily enlarge the feature from $N$ to $M + \alpha M$ representation space, as shown on the left of Fig. 2. In particular, such modification on a certain layer would enlarge the dimension of the output features, making it mismatch the following operations. Accordingly, we also extend the original kernel from $N$ to $N + \alpha N$ along the dimension, such that the original kernel size becomes $(M + \alpha M) \times (N + \alpha N) \times 3 \times 3$. Notably, the first layer of the entire network always takes 3 dimension as input (*i.e.,* RGB). Once the newly initialized filters are incorporated into the original network, all parameters are updated by the back-propagation. As training goes by, $\alpha$ linearly goes up every $n$ iterations *i.e.,* the capacity of all layers in discriminator grows up simultaneously ($n = 1$ in practice). In practice, we start with the half capacity of a standard discriminator and ensure the ending capacity is identical to the original one for a fair comparison.

**Decreasing capacity.** If the bi-classification task is relatively simple, a normal discriminator could also over-fit, which appears to memorize the training set. The synthesis quality would be thus deteriorated significantly. To mitigate this, we randomly eliminate a set of filters thus the layer width gradually shrinks, as shown on the right of Fig. 2. We explicitly control the capacity through a shrinking coefficient $\beta$. Concretely, given a certain $\beta$, we would always randomly sample a sub-kernel with $\beta M \times \beta N \times 3 \times 3$ from the aforementioned convolution layer during a certain training iteration. Different from increasing capacity, we empirically find decreasing all layers makes training unstable, especially when adjusting the lower level layers which typically contain fewer kernels. Therefore, we apply such decreasing scheme after multiple layers. Such decreasing scheme differs from the standard Dropout [42] since our method forms a "weight-level" dropout which are shared by all instances within a training batch while Dropout is more like a per-instance regularizer at "feature-level".

During training, $\beta$ also linearly goes down, leading to a discriminator with decreasing capacity. It is noteworthy that such strategy not only shrinks the network width but also to some extent introduces multiple discriminators via randomly sampling. The analysis in *Supplementary Material* demonstrates that representations derived from various discriminators could complement each other, preventing severely memorizing a certain pattern *i.e.,* alleviating the over-fitting issues substantially.

### 3.3 Two schemes for different data regimes

Prior work suggests that limited training data leads to the over-fitting of discriminator while the enhanced discriminator could also benefit from the sufficient training samples. With two basic dynamic strategies, we thus consider the bi-classification tasks under different data regimes.

**Sufficient data.** Intuitively, distinguishing the observed data from the early synthesis which is likely to be a noise is obviously much easier than the realistic synthesis at the end of training. The later stage of training thus requires a larger discriminator. We thus take the strategy of increasing capacity on sufficient data. In particular, we find starting with a relatively smaller network (*e.g.,* from one subset of the original to the entire original network) works well. That is, the extending coefficient $\alpha$ could vary from $-0.5$ to $0.0$. Such that, the largest network at the end of training is identical to the original one, *i.e.,* no extra computation is incurred in our DynamicD, compared to the baseline approach. Sec. 4.2 demonstrates that applying decreasing capacity on sufficient data makes no improvements.

**Limited data.** Since over-fitting always appears in the later stage of the training, we adopt the decreasing capacity for limited data. To be specific, the shrinking coefficient $\beta$ could start at $1.0$ and then gradually goes down to $0.5$. Considering the aforementioned unstable issue caused by decreasing capacity for all layers, we exclude low-level layers that typically contains fewer dimensions for the decreasing strategy. More analysis is available in *Supplementary Material*. Additionally, Sec. 4.2 suggests that applying increasing capacity on limited data could further exacerbate over-fitting.

**Training efficiency.** Regardless of the data regimes and adjusting strategy, the proposed DynamicD always requires less computational overhead since the largest one (*i.e.,* networks at the beginning and the end of training for decreasing and increasing respectively) is identical to the original. Therefore, DynamicD could substantially improve the training efficiency and synthesis quality. Additionally, DynamicD is agnostic to neural architecture and can be easily incorporated in other GAN training.

## 4 Experiments

We evaluate the proposed DynamicD on various synthesis tasks, across multiple datasets and under various data regimes. The experimental details are first introduced in Sec. 4.1. Sec. 4.2 contains an empirical study of two strategies under different data regimes. Sec. 4.3 reports the comparisons against prior approaches on FFHQ [29]. Lastly, the experimental results in Sec. 4.4 substantiate the generalization across multiple datasets and the synergy between DynamicD and prior techniques.

### 4.1 Setup

**Datasets.** In this work, several benchmarks are included to evaluate the proposed DynamicD from various perspectives. For instance, on FFHQ [29] which includes 70,000 high-resolution face images, we conduct the empirical study and comparison against prior approaches. In order to study the effect of different data regimes, we also follow ADA [30] to randomly sample a subset to set up a limited setting and double the entire dataset via horizontal flip for sufficient data, with all the images well aligned and cropped [33]. In addition, AFHQ-v2 [10] is also used to evaluate our DynamicD under low-data regime. To be specific, AFHQ-v2 [10] consists of around 5,000 images for dogs, cats and wild life respectively. Moreover, we conduct experiments on three sufficient scene collections *i.e.,* LSUN [58] outdoor church, bridge and bedroom which contains 126K, 818K, and 3M unique images respectively. Notably, we resize the images in FFHQ [29] and LSUN [58] to $256 \times 256$ and the images in AFHQ-v2 [10] to $512 \times 512$. Besides, we also conduct 3D-aware image synthesis on a synthetic car dataset Carla [12] containing 10,000 images rendered from 16 different car models.

**Evaluation metrics.** Akin to prior approaches, Fréchet Inception Distance (FID) [21] serves as the quantitative metric, which could reflect the human perception to some extent. Notably, in this paper, FID is usually calculated between 50,000 synthesized images and the entire training set regardless of data regimes. In particular, akin to [31], we calculate FID on 50,000 real images for LSUN [58] bridge and bedroom. The official pre-trained Inception works as the feature extractor.

**Baselines.** StyleGAN2 [31] without adaptive discriminator augmentation (ADA) [30] serves as our main baseline for 2D image synthesis. We additionally conduct 3D-aware image synthesis experiments using StyleNeRF [17]. All training settings strictly follow the prior arts to ensure the fair comparison.

Table 1: **Empirical study** on training GANs with a *capacity-varying* discriminator. All experiments are conducted on FFHQ [29] under 256 resolution, and the first row reports the number of samples used for training. We choose StyleGAN2 [31] as the baseline model, while "baseline-half" means the discriminator employs a half-width structure compared to the original, *i.e.*, "baseline-full". FID [21] (lower is better) is used to evaluate the synthesis performance. We can tell that, with a proper varying strategy, a *dynamic discriminator* substantially improves the generator capability.

| | $0.1K$ | $2K$ | $140K$ |
|---|---|---|---|
| *Fixed Capacity* | | | |
| baseline-full | 179.21 | 78.82 | 3.75 |
| baseline-half | 137.31 | 63.36 | 4.73 |
| *Varying Capacity* | | | |
| baseline-half $\rightarrow$ baseline-full | 181.03 | 63.16 | **3.53** |
| baseline-full $\rightarrow$ baseline-half | **50.37** | **23.47** | 3.74 |

## 4.2 Empirical studies

We conduct an empirical study of two proposed dynamic strategies under various data regimes introduced in Sec. 3.3. Since previous literature [64, 53, 62, 30, 25, 57, 34] usually explore the effect of data scale on FFHQ [29], we set up different data regimes of FFHQ [29] for a better comparison. To be specific, we randomly sample $0.1K$ and $2K$ images for the limited setting and augment the entire dataset via horizontal flip to build a sufficient collection with $140K$ images. With such a benchmark, we compare the dynamic strategies against two baselines. The original discriminator works as the baseline with full capacity (baseline-full) while we also directly reduce the capacity by half as a reference (baseline-half). That is, there is no dynamic adjustment of the capacity but the decreased one throughout the entire training. We implement our DynamicD with two strategies. In terms of increasing strategy, the extending coefficient $\alpha$ varies from $-0.5$ to $0.0$ such that the discriminator could be changed from the half to the full capacity. Meanwhile, we also decrease the capacity in turn via the shrinking coefficient $\beta$. Notably, all experiments make no modifications on the generator side. Tab. 1 presents the comparison of these methods.

**Varying capacity required for different data regimes.** Given a sufficient training collection with $140K$ images, a half of discriminator would lead to the poor synthesis quality, compared to the original one (4.73 *v.s* 3.75). On the contrary, a smaller network could improve the FID under low-data regimes, from 179.21 to 137.31, 78.82 to 63.36 with $0.1K$ and $2K$ samples respectively. These results also match the finding [30, 41] that reducing learnable parameters is of benefit to the limited data synthesis. It also supports the adoption of different dynamic strategies since the needed capacity varies under different data regimes.

**On-the-fly adjustment outperforming offline adjustment.** Experiments are first conducted by applying two strategies under various data regimes. According to the numbers in Tab. 1, we find that dynamically decreasing the capacity of discriminator could substantially improve the synthesis quality under low-data regimes, outperforming the fixed discriminator (even the smaller one) by a clear margin: 179.21 *v.s* 50.37 on $0.1K$, 78.82 *v.s* 23.47 on $2K$. In addition, increasing strategy from a subnet to the full network could also enhance the sufficient data generation. That is, compared to the baseline-full, our increasing strategy could achieve better FID (3.75 *v.s* 3.53) with less computational complexity throughout the entire training. These numbers demonstrate the effectiveness and superiority of the *dynamic discriminator* over the *fixed adjustment* of capacity.

**Two strategies regarding data regimes.** Although the on-the-fly adjustment of capacity could bring significant gains, the directions in varying also matter, especially under different data regimes. Tab. 1 also suggests that wrong strategy of varying capacity makes no improvements. For instance, increasing capacity hardly helps the limited data synthesis, compared to the baseline. One possible reason is that an increasing number of parameters usually exacerbate the issue of over-fitting. Another interesting finding is that, even if we reduce the capacity by half for sufficient data, FID keeps at the similar level (3.75 *v.s* 3.74). It might imply that there are a plenty of redundant parameters in the original discriminator. This intuitively answers why our DynamicD could win over the baseline "even with less computation". That is, the increasing strategy might help ensure the sufficient training of discriminator to some extent.

Table 2: **Comparison with existing approaches** that improve GANs from the discriminator side. All experiments are conducted on FFHQ [29] under 256 resolution based on StyleGAN2 [31]. FID [21] (lower is better) is reported. Our DynamicD improves GAN training from a different perspective (*i.e.*, dynamically varying the discriminator capacity) and hence is orthogonal to prior arts. The compatibility between DynamicD and other methods is explored in Sec. 4.4 and Tab. 5.Note that numbers with ∗ are obtained by our implementation.

| | $0.1K$ | $2K$ | $140K$ |
|---|---|---|---|
| DiffAugment [62] | 61.91∗ | 24.32 | 4.84∗ |
| ADA [30] | 82.17 | 15.62 | 3.88 |
| APA [25] | 65.31 | 16.91 | 3.67 |
| Adaptive dropout [30] | 90.95∗ | 67.23 | 4.16 |
| zCR [61] | 179.66 | 71.61 | 3.45 |
| InsGen [57] | 53.93 | 11.92 | **3.31** |
| Off-the-shelf pre-training [34] | - | **8.18** | - |
| StyleGAN2 [31] | 179.21 | 78.89 | 3.75 |
| StyleGAN2 [31] + DynamicD | **50.37** | 23.47 | 3.53 |

Table 3: **Generalization of DynamicD on various datasets**. FID [21] (lower is better) is reported to evaluate the synthesis performance. Note that we treat AFHQ [10] (cat, dog, wild) and LSUN [58] (church, bridge, bedroom) as *limited* and *sufficient* training settings, and hence adopt the *decreasing capacity* and *increasing capacity* schemes, respectively.

| Methods | Cat-$5K$ | Dog-$5K$ | Wild-$5K$ | Church-$126K$ | Bridge-$818K$ | Bedroom-$3M$ |
|---|---|---|---|---|---|---|
| StyleGAN2 [31] | 6.36 | 18.93 | 3.80 | 4.44 | 6.20 | 5.65 |
| *w/* DynamicD | **5.41** | **16.00** | **3.34** | **3.87** | **5.33** | **4.01** |

## 4.3 Comparison with existing approaches

In this part, we compare our DynamicD against prior approaches on both limited and sufficient data settings. StyleGAN2 [31] used in ADA [30] serves as our baseline. In addition, we include several data augmentation methods which aim at alleviating the over-fitting issues: ADA [30], APA [25] and DiffAugment [62]. Moreover, we transcribe the numbers of adaptive dropout variant from ADA [30], which implement the model augmentation *i.e.,* Dropout [50] in an adaptive manner. Moreover, we include several techniques that propose a new regularization (zCR [61]) or an extra task (InsGen [57]), and leverage pre-trained models (Off-the-shelf Models [34]) to improve GAN training respectively. It is noted that both InsGen [57] and Off-the-shelf Models [34] are based on data augmentation, making the comparison not so strictly fair to some extent. Unless specified, all methods are trained with the same iterations and architectures.

**Main results.** Tab. 2 presents the quantitative results. Our DynamicD brings the consistent improvements under all data regimes. In term of sufficient data, the proposed approach continues to improve the synthesis quality despite that the data augmentation *i.e.,* ADA [30] and model augmentation *i.e.,* Dropout [50] lead to negative impact in turn. When it comes to the limited data setting (*e.g.,* $2K$ images), DynamicD slightly outperforms DiffAugment [62] which uses a fixed data augmentation and performs worse than the adaptive ones (*i.e.,* ADA [30] and APA [25]). When there are very few training samples, like only 100 images, DynamicD beats all data augmentation methods by a clear margin. This indicates the potential of DynamicD for image synthesis under extremely limited data.

When compared against the recent techniques like zCR [61], InsGen [57] and Off-the-shelf Models [34] on the sufficient data, DynamicD achieves *competitive performances* but with *less computations and higher training efficiency*. In more details, zCR and InsGen requires extra computations across different paired images while Off-the-shelf Models [34] needs to leverage multiple pre-trained models. Unlike these approaches, our DynamicD merely increases capacity from a subnet to the normal one. More importantly, the proposed DynamicD reaches the new state-of-the-art results on extremely limited setting, outperforming InsGen [57] (50.37 *v.s* 53.93).

Table 4: **Generalization of DynamicD to 3D-aware image synthesis**. FID [21] (lower is better) is reported to evaluate the synthesis performance. We find that, for 3D-aware image generation, even the full set of FFHQ [29] and Carla [12] is insufficient for such a challenging task. Therefore, all experiments adopt the *decreasing capacity* scheme.

| Methods | FFHQ-$2K$ | FFHQ-$140K$ | Carla-$2K$ | Carla-$10K$ |
|---|---|---|---|---|
| StyleNeRF [17] | 73.50 | 8.13 | 72.1 | 53.87 |
| *w/* DynamicD | **23.29** | **7.60** | **51.0** | **47.42** |

Table 5: **Compatibility of DynamicD with existing approaches** that improve the discriminator of GANs. All experiments are conducted on 256 resolution and use StyleGAN2 [31] as the baseline model. FID [21] (lower is better) and KID [4] (lower is better) are reported as the evaluation metrics.

(a) Training on FFHQ [29].

| Methods | $0.1K$ | $2K$ |
|---|---|---|
| ADA [30] | 82.17 | 15.62 |
| *w/* DynamicD | **62.30** | **14.56** |
| zCR [61] | 179.66 | 71.61 |
| *w/* DynamicD | **66.01** | **21.08** |

(b) Fine-tuning on MetFaces [30].

| Methods | FID | KID ($\times 10^3$) |
|---|---|---|
| Fine-tuning | 22.93 | 5.17 |
| *w/* FreezeD [41] | 22.15 | 4.33 |
| *w/* DynamicD | **20.52** | **2.39** |

## 4.4 Generalizability and compatibility of DynamicD

In this part, we first verify the generalizability of the proposed DynamicD across various datasets and tasks, and then study its compatibility with existing discriminator-improving techniques.

**Generalization across datasets.** We choose AFHQ-v2 [10] and LSUN [58] as the evaluation benchmarks because of their data regimes. StyleGAN2 [31] used in [30] serves as our baseline. Tab. 3 and Fig. 3 present the quantitative and qualitative results respectively.

The synthesis performances are substantially improved given both limited and sufficient data. Decreasing capacity in AFHQ-v2 [10] boosts the FID on cat, dog and wild life domains respectively. Importantly, as the data regimes scale up, our DynamicD could *improve training efficiency and bring substantial gains simultaneously*. In particular, the gain becomes larger when increasing the training samples from $818K$ (bridge) to $3M$ (bedroom), implying the potential of DynamicD in large-scale content generation (*e.g.,* training a GAN on ImageNet [11]).

**Generalization across tasks.** Going beyond 2D image synthesis, we also apply our DynamicD on popular 3D-aware image generation [48, 43, 8, 17, 56]. It aims at producing realistic images with high multi-view consistency, by incorporating implicit functions or differentiable rendering into generators. We take StyleNeRF [17] as an example, which uses the same discriminator of StyleGANv2 [31]. Considering there lacks baselines of 3D GANs under low-data regimes, we follow ADA [30] to randomly sample a subset out of the entire collection. Tab. 4 shows the quantitative results.

We can see that limited data indeed leads to poor quality of 3D-aware synthesis. Besides, we empirically find decreasing capacity works better on full set of both FFHQ [29] and Carla [12]. Thus our DynamicD can be also used for improving 3D-aware image synthesis.

**Compatibility with discriminator-improving techniques.** We have demonstrated the effectiveness of our approach. It would be even better if adjusting capacity is compatible with previous methods of improving discriminator from various perspectives. For instance, ADA [30] and zCR [61] are proposed to improve the data efficiency and training stabilization respectively. We thus conduct the compatibility experiments under low-data regimes on FFHQ [29]. Tab. 5a provides the results. Obviously, equipped with DynamicD, these approaches could enjoy the consistent improvements.

Moreover, as prior literature [34, 47, 41] shows that leveraging pre-trained models in discriminator could help training and data efficiency, we also wonder if the pre-training is compatible with DynamicD. Meanwhile, considering using frozen one might make the metrics unreliable, we thus choose the generative domain adaptation task as a benchmark. It basically fine-tunes a given pre-trained model which is usually trained on a large-scale source domain (*e.g.,* FFHQ [29]) on a target domain. Concretely, we first pre-train a StyleGAN2 on FFHQ [29] without any modifications of capacity and then fine-tune this model with DynamicD on the target domain MetFaces [30] which contains around 1336 high-quality faces collected from an art collection. Note that all images of

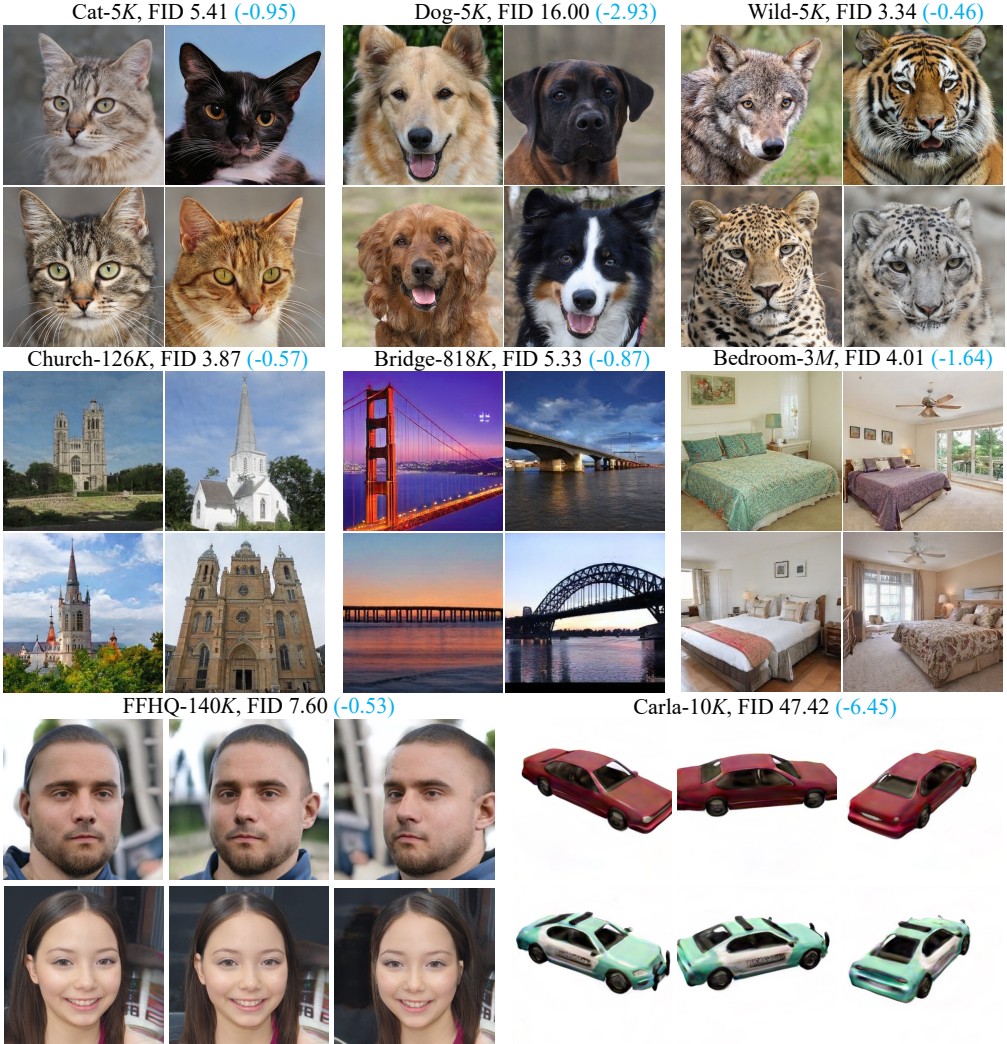

Cat-5*K*, FID 5.41 (-0.95)   Dog-5*K*, FID 16.00 (-2.93)   Wild-5*K*, FID 3.34 (-0.46)

Church-126*K*, FID 3.87 (-0.57)  Bridge-818*K*, FID 5.33 (-0.87)  Bedroom-3*M*, FID 4.01 (-1.64)

FFHQ-140*K*, FID 7.60 (-0.53)   Carla-10*K*, FID 47.42 (-6.45)

Figure 3: **Qualitative results** on various datasets. Dataset scale and FID are listed above. Numbers in **blue** highlight the improvements over baselines.

MetFaces [30] are resized to $256 \times 256$ resolution. Tab. 5b presents the FID and kernel inception distance (KID) [4], demonstrating the compatibility of the proposed approach.

## 5 Conclusion

We propose a general method DynamicD for improving GANs. By adjusting capacity of discriminator under two different schemes, we can substantially enhance image synthesis quality and reduce the computational cost accordingly. Experiments on a wide range of datasets and generation tasks demonstrate the effectiveness, generalizability, and compatibility of our DynamicD, with the consistent performance gains.

**Discussion.** Despite the appealing synthesis quality and performances across various tasks and datasets, our DynamicD still has some limitations. For instance, current form of DynamicD adjusts network capacity by extending or shrinking layer width. It is not explored for the influence of other factors such as network depth. Meanwhile, current experiments are conducted on CNN-based discriminator. Gains on transformer-based discriminator [26] remain uncertain and valuable to investigate. On the other hand, although this work makes an early attempt to demonstrate the effectiveness of two dynamic schemes under various data scales, some self-adjusting or AutoML strategy which might be more effective. Moreover, a theoretical study would make it more appealing, left for future study.

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
