# Improving GANs with A Dynamic Discriminator
## – *Supplementary Material* –

This supplementary material is organized as follows. We first discuss the broader impact of the proposed DynamicD in Sec. A. More implementation details are provided in Sec. B to ensure the reproduction. Addtionally, we present the analysis of various sub-nets in Sec. C and more qualitative results in Fig. S2 and Fig. S3 respectively. Besides, Sec. D presents the training dynamics for the further analysis. Sec. E also conducts qualitative experiments to verify whether our approach memorizes the real images for extremely limited data. At last, Sec. F shows the hyper-parameter analysis.

## A   Broader impact

Obtaining appealing synthesis quality is a fundamental and practical problem. The proposed approach has no doubt substantially advancing the field of image synthesis under various data regimes. It demonstrates the importance of discriminator in the two-player competition as simply adjusting the capacity could lead to such significant improvements on a variety of settings, making training generative models more accessible to everyone. On the other hand, this technique would reduce the cost of fake content synthesis like Deepfake which might generate negative societal impact. We strongly oppose the abuse of our method in violating any security and privacy issues, and we believe such negative impact will be lowered with the rapid development of deep fake detection technique.

## B   Implementation details

DynamicD is developed based on the official implementation of StyleGAN2-ADA. We keep the same architecture of generator, training regularization, hyper-parameters, optimizers and loss functions to ensure the fair comparison. For 2D image synthesis, we use the configuration recommended by ADA [2] on various datasets. In order to speed up the training, we follow ADA [2] to use mixed-precision training on a server with 8 GPUs. Notably, the total number of seen images for sufficient data training is 25 million regardless of datasets. In terms of 3D-aware image synthesis, we implement DynamicD based on the official code of StyleNeRF but disable the progressive growing training in practice.

## C   Analysis on various sub-nets

As mentioned in Sec.3, our decreasing strategy would involve multiple sub-nets. In order to analyze the behaviors of various sub-nets, we leverage the Grad-CAM [4] to visualize the spatial attentions. Fig. S1 presents the attention maps of baseline and our approach on FFHQ-$2K$ [1]. Note that we observe and compare the visual attention of different approaches at one training step.

Given a real image, the fixed discriminator would prefer to a certain spatial location. That is, the spatial focus largely determines whether this image is recognized as real or fake one. Nevertheless, the subset randomly sampled from a discriminator with decreasing capacity pours attention differently. The average attention maps derived from ten sub-nets indicate that such various networks could complement each other to some extent, helping discriminator look at more regions to make a decision.

## D Training dynamics

In this section, we plot the time-varying FID curves under several settings for further analysis. First, performance curves under two limited and two sufficient datasets are present in Fig. S4. Obviously, at the very start of training, our approach has little advantage since not all the parameters get trained due to the subnet sampling. But after a short period of warming up, our approach outperforms the baseline consistently. To some extent, we could tell that our approach could speed up convergence slightly since it usually takes less time for our approach to reaching the same FID.

Moreover, we also present the FID and logits of the discriminator (*i.e.,* the output of discriminator) in generative domain adaptation task in Fig. S7. Obviously, FreezeD [3] and baseline (*i.e.,* directly fine-tuning) seem to overfit since their FIDs gradually goes up, with increasingly higher confidences in real/fake bi-classification. On the contrary, as the capacity of the discrimanator is progressively limited, the overfitting is alleviated to some extent.

As mentioned in Sec. 4.4, we empirically find decreasing capacity works better on full set of both FFHQ. Here, we plot the FID curves of StyleNeRF baseline, StyleNeRF with increasing capacity and StyleNeRF with decreasing capacity respectively in Fig. S8. It suggests that 70,000 face images seem also insufficient for 3D-aware image synthesis since decreasing strategy is always better than StyleNeRF baseline and increasing scheme.

## E Memorization verification

For extremely limited setting like 100 training samples, memorization of real images usually occurs. In order to verify the memorization, we first conduct latent interpolation visualization in Fig. S5. Moreover, we also follow [5] to perform the nearest neighbor test in Fig. S6. To be specific, multiple synthesized images serve as queries to retrieve the most similar real images of the training set. Here, the similarity between query and real images is measured by pixel-wise $\mathcal{L}_1$ distance. Qualitative results demonstrate that our approach does not directly memorize the real images.

## F Hyper-parameters analysis

Tab. S1 present the analysis of multiple hyper-parameters on FFHQ. Obviously, our approach is not highly sensitive to different hyper-parameters like decreasing schedules, how many layers are supposed to be excluded from decreasing channels, and decreasing/increasing coefficients. In particular, the hyper-parameters chosen on FFHQ keep identical across various datasets and tasks.

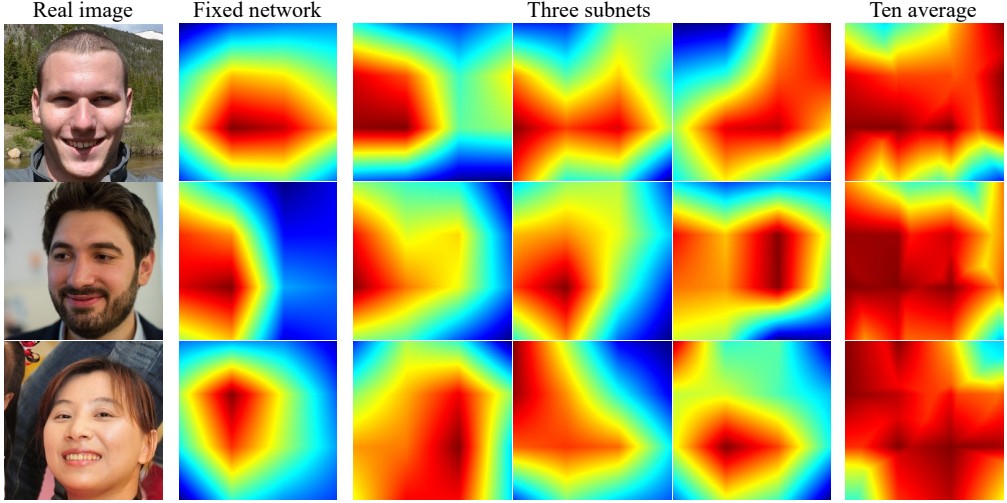

Figure S1: **Attention maps** produced by (1) a *fixed* discriminator in training, (2) three sub-nets sampled from our *dynamic* discriminator, and (3) averaging ten sub-nets. We can tell that our DynamicD covers more local regions when making the real/fake decision.

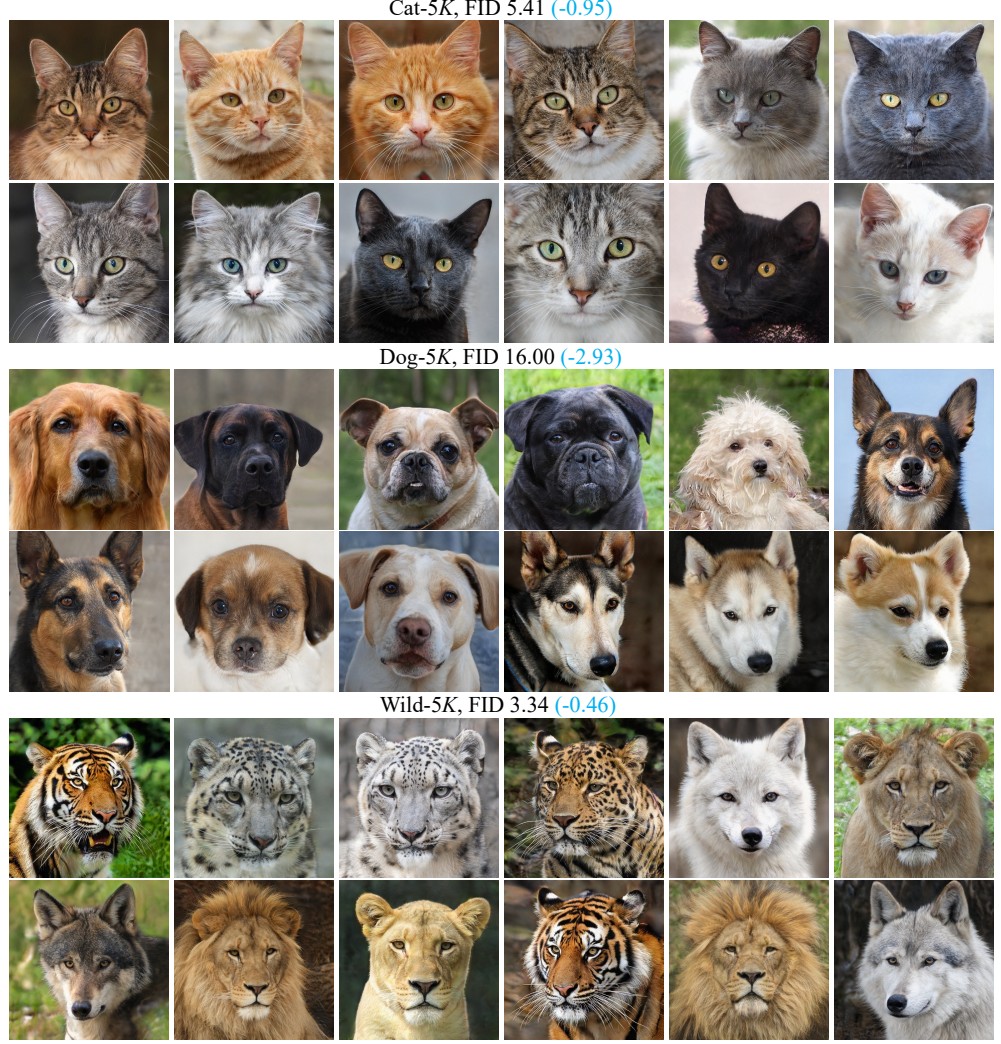

Figure S2: **Qualitative results** on various datasets. Dataset scale and FID are listed above. Numbers in **blue** highlight the improvements over baselines.

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

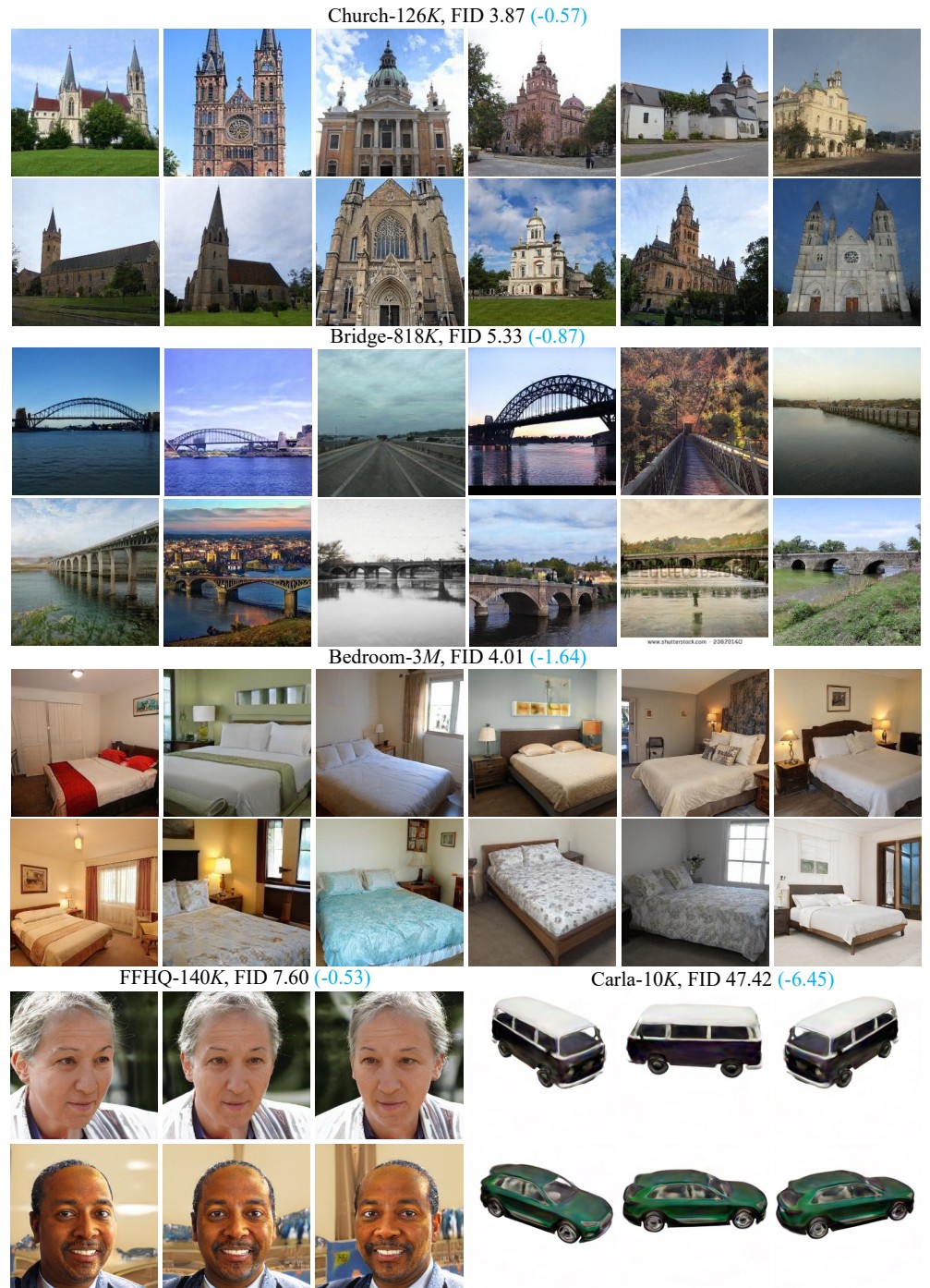

Figure S3: **Qualitative results** on various datasets. Dataset scale and FID are listed above. Numbers in **blue** highlight the improvements over baselines.

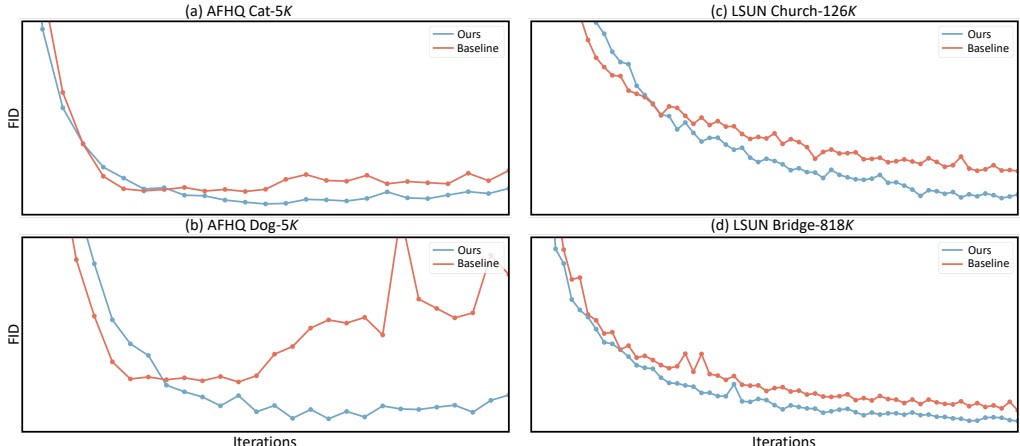

Figure S4: **Time-varying FID curves** on various datasets under both limited (*i.e.*, (a) and (b)) and sufficient data (*i.e.*, (c) and (d)). Obviously, our method constantly outperforms baseline approach after a short period of warming up.

Latent Interpolation

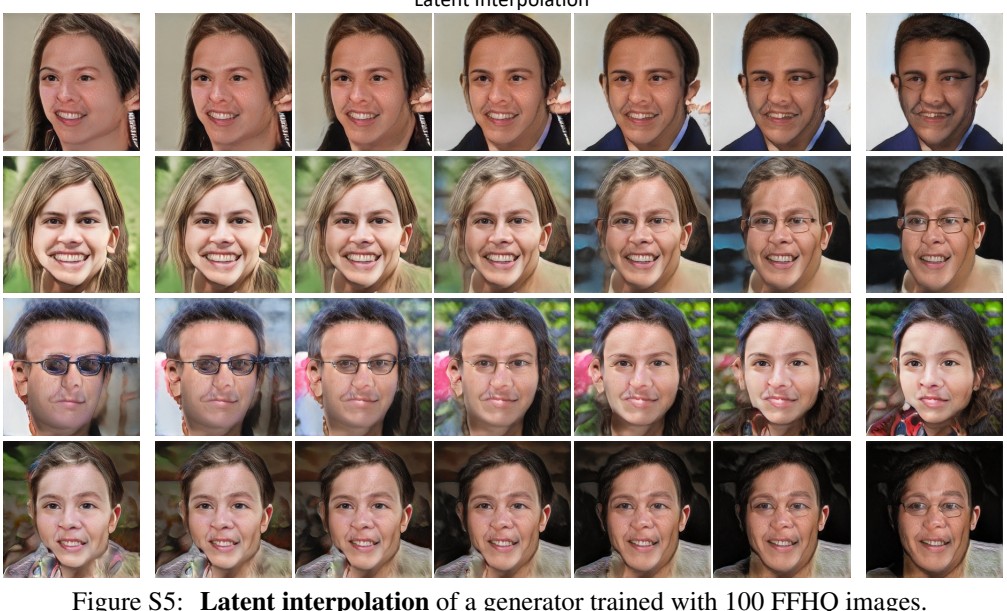

Figure S5: **Latent interpolation** of a generator trained with 100 FFHQ images.

Synthesis    Nearest Neighbors                    Synthesis    Nearest Neighbors

Figure S6: **Nearest neighbors** in pixel space. Each synthesis serves as a query to retrieve the most similar real images of collection. Note that the similarity is measured by pixel-wise $\mathcal{L}_1$ distance.

Table S1: **Hyper-parameters analysis** conducted on FFHQ. (a) analyzes the effect of different decreasing schedules and (b) studies how many low-level layers are supposed to be excluded from decreasing channels. Besides, (c) and (d) investigates the increasing/decreasing coefficients. Note that settings with $*$ are chosen as the default ones for all datasets and tasks.

<table>
<tr><td colspan="2">(a) Different schedules</td><td colspan="2">(b) #layers excluded from decreasing channels</td></tr>
<tr><td>FFHQ-2$K$</td><td>FID</td><td>FFHQ-2$K$</td><td>FID</td></tr>
<tr><td>Baseline</td><td>78.89</td><td>Baseline</td><td>78.89</td></tr>
<tr><td>linear-decreasing$^*$</td><td>23.47</td><td>6 layers</td><td>29.05</td></tr>
<tr><td>cosine-decreasing</td><td>22.67</td><td>9 layers$^*$</td><td>23.47</td></tr>
<tr><td></td><td></td><td>12 layers</td><td>30.38</td></tr>
</table>

(c) **Decreasing capacity.** Numbers on the sides of $\rightarrow$ denotes the network capacities in the beginning and the end respectively

(d) **Increasing capacity.** Numbers on the sides of $\rightarrow$ denotes the network capacities in the beginning and the end respectively

<table>
<tr><td>FFHQ-2$K$</td><td>FID</td><td>FFHQ-140$K$</td><td>FID</td></tr>
<tr><td>Baseline</td><td>78.89</td><td>Baseline</td><td>3.75</td></tr>
<tr><td>$1.0 \rightarrow 0.7$</td><td>30.37</td><td>$0.7 \rightarrow 1.0$</td><td>3.62</td></tr>
<tr><td>$1.0 \rightarrow 0.5^*$</td><td>23.47</td><td>$0.5 \rightarrow 1.0^*$</td><td>3.53</td></tr>
<tr><td>$1.0 \rightarrow 0.3$</td><td>30.32</td><td>$0.3 \rightarrow 1.0$</td><td>3.87</td></tr>
</table>

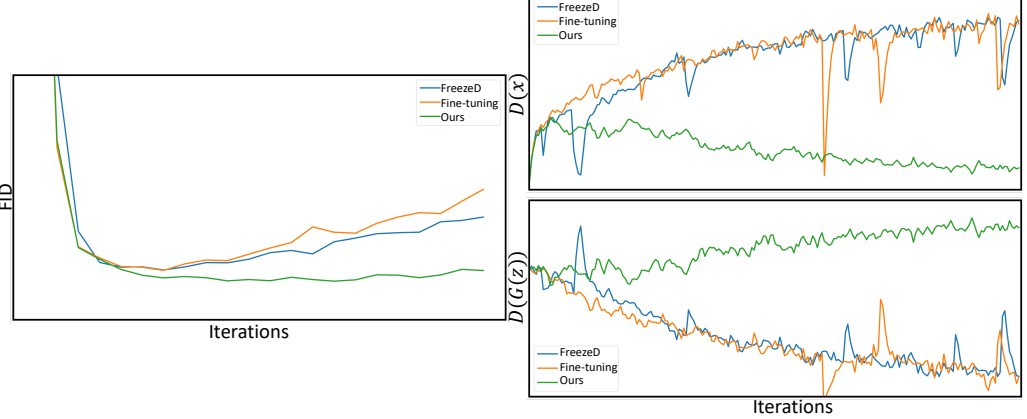

Figure S7: **Generative domain adaptation analysis.** FID curve suggests that our method constantly outperforms other approaches. In terms of distinguishing real images $\mathbf{x}$ from the synthesized ones $G(\mathbf{z})$, logit of discriminator also shows that other approaches have higher confidence than ours, indicating the risk of overfitting to some extent.

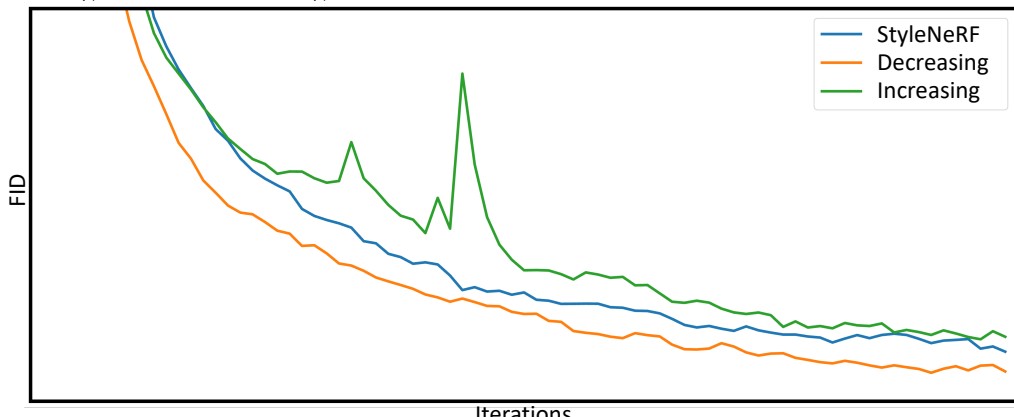

Figure S8: **FID curves of StyleNeRF** might imply that 3D-aware image synthesis requires data from more than quantity and diversity perspectives (*e.g.,* multi-view observations) since decreasing strategy is always better than StyleNeRF baseline and increasing scheme *i.e.,* 70,000 face images seem insufficient for 3D-aware image synthesis.

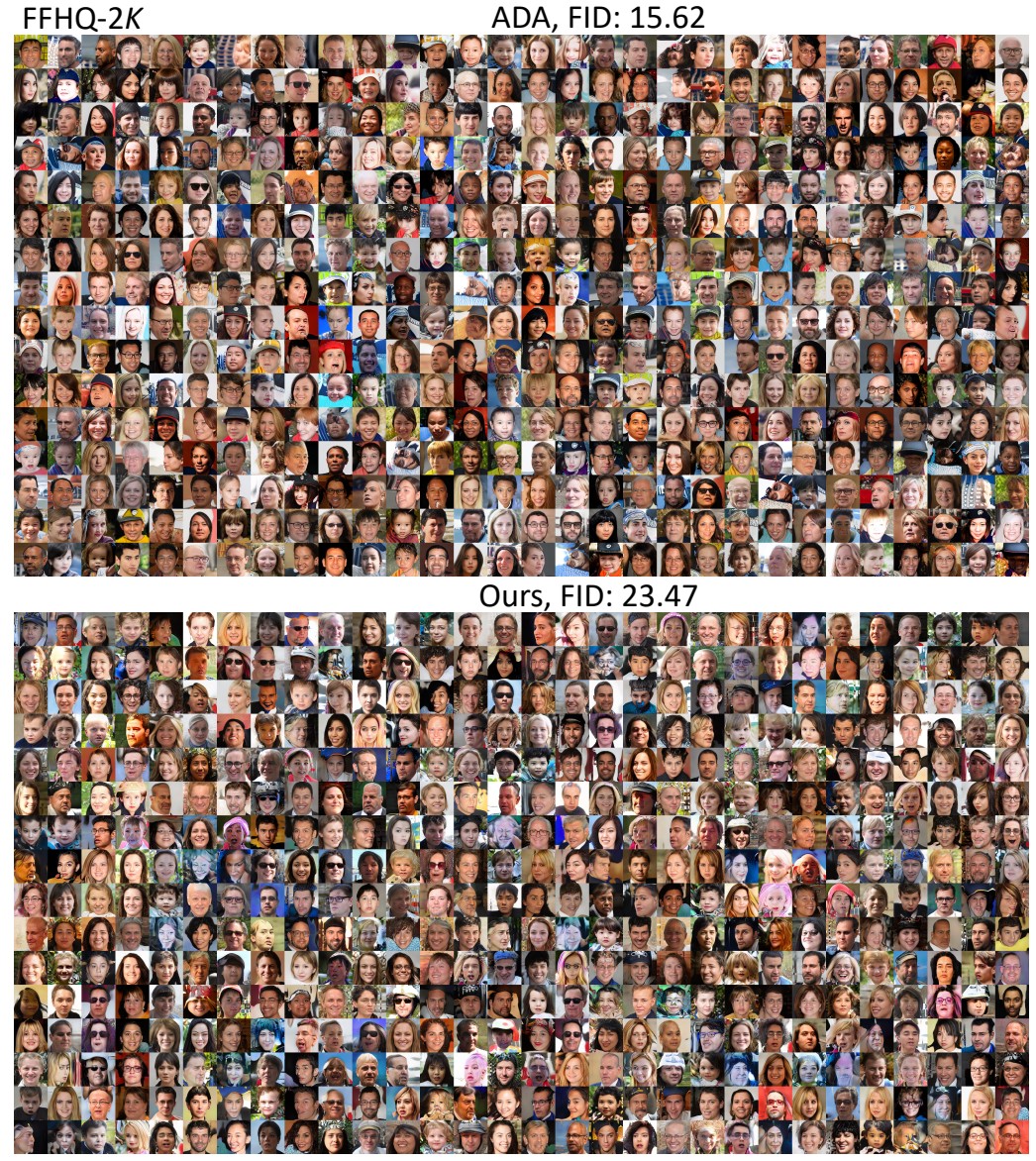

Figure S9: **Uncurated images** synthesized by ADA and our method (trained on FFHQ-2$K$). Zoom in for better view.