# OpenReview forum: "Improving GANs with A Dynamic Discriminator"
_NeurIPS.cc/2022/Conference — NeurIPS 2022 Accept_

### Official Review · Reviewer_h4W2 · 2022-07-10

**Rating:** 6
**Confidence:** 4
**Soundness:** 3 good
**Presentation:** 3 good
**Contribution:** 2 fair

**Summary:**

The paper proposes a simple method for avoiding overfitting and underfitting in the discriminator network of a GAN. This work proposes linearly adjusting the capacity of the discriminator network during training in order to deal with these problems. To deal with underfitting, the authors propose gradually increasing the number of feature maps of each layer. In cases where data is insufficient and the discriminator is prone to overfitting the authors propose gradually subsampling feature maps from each layer, effectively reducing its capacity. Results are presented on image synthesis and 3D-aware image synthesis and compared to StyleGAN2 and StyleNerf baselines. Furthermore, the proposed approach is compared and also used in combination with other methods that avoid discriminator overfitting.

**Questions:**

- Although the starting and ending values α and β are reported in the paper the number of iterations n after which their values are adjusted is not stated.
- The supplementary material mentions that "Notably, the total number of seen images for sufficient data training is 25 million regardless of datasets". How did the authors come up with this number and why is it independent of the dataset?
- I am unsure what to make of the attention maps obtained using Grad-CAM presented in the supplementary material. The authors should provide further insights as to why the discriminator focusing on the entire image is beneficial. Also to some extent, it is expected that the focus of the discriminator will change during training even at a fixed capacity. Hence averaging 10 attention maps throughout training even for the fixed discriminator should also result in a larger attention map.
- The paper mentions that the proposed approach is more efficient. It is clear that the adaptive capacity reduces the memory required at certain stages of training (at the start or end depending on strategy). However, in practical terms, what difference does this make during training and how do the authors take advantage of this? Does training take less time than other approaches?

**Limitations:**

The authors have sufficiently discussed the limitations of the work and possible future improvements. I do not believe that there are significant ethical concerns with regard to this work (beyond those associated with generative modelling in general) hence I do not believe the paper requires an additional discussion about societal impact.

**Strengths And Weaknesses:**

## Strengths:
- The paper is well motivated. The problem of overfitting discriminators is known to affect the ability of GANs to produce realistic samples with limited data.
- The proposed method is simple and can be applied to most network architectures.

## Weaknesses:

- A citation to DropoutGAN[1*] should be added to the text since it is one of the first methods to propose using dynamically changing discriminators during training.
- The novelty of the method is limited. The problem of overfitting discriminators is well known and several approaches have already suggested adaptively limiting the expressive power of the discriminator to solve it [30, 1*].
- The proposed method interpolates between 2 network capacities but does not explain how these are selected and how important their choice is. In the paper, the authors start with a "standard" full discriminator capacity but for new tasks, this is not known in advance. Furthermore, for a novel task, it is not immediately clear when to select a decreasing or increasing strategy because in practice it is hard to know in advance what constitutes "sufficient" training data. Choosing the correct strategy and capacity seem to be important hyperparameters that require additional experiments to determine.
- Compared to existing approaches in Table 2 the proposed method is only better than other approaches when dealing with tiny datasets (100 images). It seems to perform worse than nearly all other approaches when 2K images are used for training and comparably when 140K images are used.
- Line 25 states that improvements of the discriminator are "much less explored" than the generator. This is not accurate. It is well known that the discriminator drives the generator to learn the correct distribution and many different discriminator approaches have been proposed in order to improve results [24, 1*, 2*, 3*]. This statement should be revised.
- Certain details are not mentioned in the paper (see Questions section)

[1*] Dropout-GAN: Learning from a Dynamic Ensemble of Discriminators, Mordido et.al. ACM KDD 2018
[2*] Conditional Image Synthesis with Auxiliary Classifier GANs, Odena et. al. ICML 2017
[3*] cGANs with Projection Discriminator, Miyato and Koyama, ICLR 2018

---

> ### Author Response · Authors · 2022-08-02
> **Response to Reviewer h4W2 (Part 1)**
>
> Thanks for the valuable comments. Individual concerns are addressed as follows. Note that, to make sure the revision is easy to track, we do not change the paper structure, and instead list all additional results in the supplementary material. We will rearrange the materials to make them fit in 9 pages in the next version.
>
> **Q1. A citation to DropoutGAN[1\*] should be added to the text since it is one of the first methods to propose using dynamically changing discriminators during training.**
>
> A1: Thank you for pointing out this paper published at KDD. It is now cited *in the revised version (Line 98)*. It should be emphasized that our approach *clearly differs* from DropoutGAN. DropoutGAN employs a collection of discriminators and randomly select some of the discriminators to train in each iteration. In that case each discriminator is learned independently (*i.e.*, they do not share information from each other). Differently, we study the alignment between the model capacity of a single discriminator and the evolving task difficulty. We discuss the differences *in the revised version (Line 98)*.
>
> **Q2. The novelty of the method is limited. The problem of overfitting discriminators is well known and several approaches have already suggested adaptively limiting the expressive power of the discriminator to solve it [30, 1\*].**
>
> A2: These three methods all attempt to improve synthesis, but with totally different approaches. Thus, each method has its unique novelty:  [30] applies data augmentation to overcome the overfitting issue, where the expressive power of the discriminator does not change. [1*] selects a subset of discriminators from a large collection to tackle the mode collapse issue, where the expressive power of each discriminator remains the same. Our approach is motivated by the observation in the previous literature, that “a more difficult task may require a more powerful model”. This clearly differs from [30, 1*]. Our key contribution is to show GAN training benefits from a dynamic discriminator capacity, rather than the adaptive adjustment of the training process itself (which is indeed a widely used training strategy)
>
> Also, decreasing (which is relevant to [1*]) is only one strategy of our approach. Our capacity adjustment also has an increasing counterpart. This strategy targets the setting with sufficient training data, which is beyond the widely studied “overfitting issue”.
>
> **Q3. How to select the start and end capacity of the discriminator.**
>
> A3: The “standard” full discriminator denotes the original/vanilla discriminator in every baseline. For example, we use the standard StyleGAN2 discriminator as the “full” discriminator. As shown in *Tab. S1 in the revised supplementary material (Page 6)*, both using 0.5 capacity at the start point of increasing scheme and using 0.5 capacity at the end point of decreasing scheme suggest the best results. Note that, we do *not* spend too much on tuning these hyper-parameters, where we use the same values for all our experiments. As for whether to choose the increasing or the decreasing scheme for a new dataset, only two experiments are needed to run.
>
> **Q4. Comparable results to existing approaches in Tab. 2.**
>
> A4: Different from the alternatives in Tab. 2, this work offers a **new direction** to improve GAN training, which is to vary the network capacity of the discriminator. It is *orthogonal and synergistic* to many existing data augmentation approaches, as validated in Tab. 5. We believe that our discovery could help the community with a better understanding on GAN training, and encourage more research in this direction.
>
> Furthermore, the results in Tab. 2 have already demonstrated the superiority of our approach over our direct baseline (*i.e.*, StyleGAN2). The results on various datasets (*e.g.*, AFHQ and LSUN) and tasks (*e.g.*, 2D image synthesis and 3D-aware image synthesis) verify the effectiveness and generalizability of our method.

---

> > ### Author Response · Authors · 2022-08-02
> > **Response to Reviewer h4W2 (Part 2)**
> >
> > **Q5. Statement about “much less explored”**
> >
> > A5: What we want to say is that the studies of discriminators are relatively less studied than those of generators. We rephrase this sentence in the revised version (Line 26)*.
> >
> > **Q6. Although the starting and ending values α and β are reported in the paper the number of iterations n after which their values are adjusted is not stated.**
> >
> > A6: We use $n=1$ in practice, which is updated *in the revised version (Line 147)*.
> >
> > **Q7. How did the authors come up with the total number of seen images and why is it independent of the dataset?**
> >
> > A7: We follow the original StlyeGAN2 and StlyeGAN2-ADA papers, which usually train a GAN until the discriminator has seen 25 million real images.
> >
> > **Q8. The authors should provide further insights as to why the discriminator focusing on the entire image is beneficial. Averaging 10 attention maps throughout training even for the fixed discriminator should also result in a larger attention map.**
> >
> > A8: First, image synthesis requires the generator to produce all details satisfyingly. Hence, a discriminator with a focus on a larger region would help the generator in such a task.
> >
> > Second, please note that the 10 attention maps are obtained from different subnets **at the same training step**. We clarify this *in the revised supplementary material (Line 29)*.
> >
> > **Q9. About training efficiency.**
> >
> > A9: Since we are always using a subnet of the discriminator for training, the computational cost is always less than the baseline. Under our default setting (*i.e.*, using half channels at the start of increasing strategy or at the end of decreasing strategy), it requires 15.97 hours when using 8 Nvidia A100 GPUs for training, while the baseline requires 18.75 hours. Our approach is around **15% faster**.
> >
> > Besides, we also plot the time-varying FID curves *in Fig. S4 of the revised supplementary material (Page 5)*, where our method outperforms the baseline consistently.

---

> > ### Comment · Reviewer_h4W2 · 2022-08-07
> > **Regarding start and end capacity of the discriminator**
> >
> > I would like to thank the authors for their responses, which have provided clarification on many of the points raised in the reviews. The question of the robustness to hyperparameters and training schemes was brought up by more than one reviewer so I would like some further clarification on the tuning required. The authors mentioned in the rebuttal that the model is robust with regard to the choice of additional hyperparameters α, and β but do not comment on the robustness to the initial and final capacities. In their rebuttal, the authors mention that the full capacity is just the "standard" and that only two experiments are required in order to select the increasing and decreasing scheme. However, these capacities are only "standard" and known in advance because someone else has performed rigorous fine-tuning to obtain the best possible model and report in a paper.  How well does the method perform if the full capacity is not taken directly from an existing paper? This is important because the authors have made the claim that the proposed approach is more efficient for training than the existing approaches. However, this is not true if for a novel problem/dataset the authors need to first find the optimal full capacity and then perform an additional 2 experiments to determine the adaptive capacity schema.

---

> > > ### Author Response · Authors · 2022-08-08
> > > **Discussion about the “standard” capacity and training efficiency.**
> > >
> > > Thank you again for your review and discussion. Here, we would like to clarify some misunderstandings.
> > >
> > > - You are correct that “it takes effort to tune the hyper-parameters to get the best possible model for a new dataset”. However, those hyper-parameters (like gradient penalty and model capacity in StyleGAN2) are **introduced by the baseline model instead of our DynamicD**. In other words, like most existing approaches on improving GAN training (*e.g.*, ADA, zCR, InsGen, *etc.*), we assume the baseline is *already well-tuned* and attempt to improve the generation performance *on top of that*. Hence, the well-tuned baseline provides the “standard” capacity, which serves as the initial/final capacity in our approach. Tuning the baseline to its optimal is not our focus. By the way, StyleGAN2 (with the official implementation [here](https://github.com/NVlabs/stylegan2)) has already provided a general model capacity that can be widely used for various datasets and achieve satisfying performance (may be not optimal but good enough). In our work, we just follow that setting and treat it as the baseline.
> > >
> > > - After choosing the baseline, our focus is to improve its performance by dynamically adjusting the discriminator capacity during training. Considering the two proposed strategies (*i.e.*, increasing and decreasing), we only need two experiments to determine which strategy would be more suitable for a given dataset. As you have already agreed, our strategies are **robust to the hyper-parameters ($\alpha$ and $\beta$) introduced in this work**.
> > >
> > > - About the claim “the proposed approach is more efficient for training than the existing approaches”. By saying more *efficient* for training, what we want to express is that “our model can be trained *faster* than the baseline because only a part of parameters are updated in each training step”. Please also refer to **Q4 to Reviewer SnfE** for the training time comparison and the convergence curves.
> > >
> > > We will also add the above discussion in the next version to help the readers with a better understanding of the scope of this work.

---

### Official Review · Reviewer_Xxwt · 2022-07-10

**Rating:** 5
**Confidence:** 4
**Soundness:** 3 good
**Presentation:** 3 good
**Contribution:** 2 fair

**Summary:**

In the training stage of GANs, the synthesized data distribution varies with the evolving generator. Thus the discriminator has a distribution shift issue. To tackle this issue, the paper proposes adjusting the discriminator's capacity on the fly (named DynamicD). The paper presents two capacity adjusting schemes and confirms that different training data regimes have different favored schemes. The paper gradually increases the network width by including newly initialized filters to increase discriminator capacity, and progressively decreases
the network width by randomly dropping a subset of filters to decrease capacity. Extensive experiments substantiate the generalizability of the DynamicD.

**Questions:**

Please see the weaknesses.

**Limitations:**

Yes, they did.

**Strengths And Weaknesses:**

Strengths:

1. The paper is well-written, and the experiments are comprehensive.

2. The problem studied in the paper is important and would be a good contribution to the community.

3. The method is simple and easy to follow.

4. The proposed method shows a large improvement in the image generation task, especially when the training dataset is limited


Weaknesses:

1. In Table 4, "we find that, for 3D-aware image generation, even the full set of FFHQ and Carla is insufficient ... all experiments adopt the decreasing capacity scheme." It seems a little counter-intuitive. FFHQ has as many as 70,000 face images, which is enough for the face manifold. Why use a decreasing capacity scheme instead of an increasing capacity scheme? If possible, the authors can show the FID curves under both schemes.

2. The increasing/decreasing coefficient is an important hyperparameter for the dynamic discriminator. The author should do ablation experiments on this coefficient in detail.

3. For generative domain adaptation experiments (L316), it is a commonly used and effective method for training GANs with limited data. In order to clearly compare the different methods, I suggest the author provide FID curves for different settings such as w/ DynamicD, w/o Dynamic, w/ DiffAug. In addition, if possible, the logits of the discriminator (i.e., D(x), D(G(z))) also help us understand the training dynamics, thus which also need to be provided.

4. L273, "DynamicD slightly outperforms DiffAugment [60] which uses a fixed data augmentation and performs worse than the adaptive ones (i.e., ADA [30] and APA [25])." In fact, ADA has the risk of leaking rotating augmentation onto the generated images under the limited data setting. Therefore, a lower FID does not mean that ADA is better than DiffAug. If possible, the author should compare more uncurated generated images of different methods to get a visual impression of the quality of the methods.

---

> ### Author Response · Authors · 2022-08-02
> **Response to Reviewer Xxwt**
>
> Thanks for the valuable comments. Individual concerns are addressed as follows. Note that, to make sure the revision is easy to track, we do not change the paper structure, and instead list all additional results in the supplementary material. We will rearrange the materials to make them fit in 9 pages in the next version.
>
> **Q1. Why does 3D-aware image generation use the decreasing scheme on FFHQ?**
>
> A1: 3D-aware image generation is a far more challenging task than 2D image synthesis, because it requires the model to capture the underlying 3D geometry from 2D images. For example, it requires the model to learn the pose information from different samples (which are 3D inconsistent across views). We believe this is the reason why 70,000 faces are enough for the 2D face manifold yet insufficient for the 3D face manifold. As suggested, we include the FID curves of StyleNeRF (baseline) and our two different strategies *in Fig. S8 of the revised supplementary material (Page 6)*. It shows that the decreasing strategy is indeed better than its increasing counterpart.
>
> **Q2. Ablation experiments on the increasing/decreasing coefficients.**
>
> A2: *Tab. S1 in the revised supplementary material (Page 6)* gives the ablation results. We observe that half capacity is the best one for both increasing and decreasing strategy. It’s noteworthy that an exhaustive tuning of the increasing/decreasing coefficients is not necessary, and we use the same coefficients in all experiments across various datasets and tasks.
>
> **Q3. Analysis (FID curves and logits) on the domain adaptation task.**
>
> A3: *Fig. S7 in the revised supplementary material (Page 6)* shows the time-varying FID and logits regarding the results in Tab. 5b. We can observe that the performances of both fine-tuning and FreezeD get worse at the later training stage. This is caused by the overfitting issue, where the discriminator gets more confidence in differentiating real/fake samples (see logits). Differently, our approach presents a more stable performance.
>
> **Q4. ADA has the risk of leaking rotating augmentation onto the generated images under the limited data setting. The author should compare more uncurated generated images of different methods to get a visual impression of the quality of the methods.**
>
> A4: Under the setting of limited training data, data augmentation indeed suffers from the risk of augmentation leak. (Notably, our approach does not have such a risk). As suggested, we include the visual results (*i.e.*, a collection of uncurated generated images) *in Fig. S9 of the revised supplementary material (Page 7)*.

---

### Official Review · Reviewer_SnfE · 2022-07-11

**Rating:** 5
**Confidence:** 5
**Soundness:** 3 good
**Presentation:** 3 good
**Contribution:** 3 good

**Summary:**

The paper proposes a dynamic discriminator scheme for training generative adversarial networks. This mitigates the overfitting issues in limited data scenarios and improve GAN training in large data settings by ensuring a balance b/w the generator and discriminator. The dynamic discriminator is achieved by increasing/decreasing the number of channels during training. The experiments conducted show the benefit of the dynamic discriminator across different settings.

**Questions:**

1. In line 294 the paper mentions improved training efficiency. It would be great if quantitative numbers can be reported in terms of training time and convergence curves.
2. Table 5 (b) MetFaces KID w/ FreezeD setting is reported as 4.33. But [30] (StyleGAN2-ADA https://arxiv.org/pdf/2006.06676.pdf) Fig 11 (a) reports similar setting KID as 2.05. Any comments on the difference? As I understood both settings are training w/ FreezeD on MetFaces from a pretrained model on FFHQ.
3. The schedule of increasing alpha or decreasing beta is not mentioned in the paper. Is it linear and does having different schedules affects training? Also, decreasing channels schedule is only applied to higher level layers. What are the specific layer numbers and how does this choice affect the results?
4. In line 262 it is mentioned that “we extract the numbers of adaptive dropout from ADA”. Not sure what is meant by this. StyleGAN2-ADA doesn’t involve any dropout in discriminator layers.
5. Minor comment: line 28 “line fundions”


**Limitations:**

Yes.

**Strengths And Weaknesses:**

Strengths:
1. The proposed method of dynamic discriminator to improve GAN training is nice, intuitive and the results show its benefit in different settings.
2. The paper is well written and is easy to understand.

Weakness:
1. Comparison of the method “full -> half” with random half channel selection at each training step from a full channel configuration is not shown. It’s not clear if the benefit from the method is due to random selection of channels which will also help in preventing overfitting issues in limited data scenario.
2. Comparison of the method with StyleGAN2+ADA (or StyleGAN2 + DiffAugment) in Table 2 and Table 3 instead of only having it as an ablation experiment. As mentioned in the supplementary, the code is built upon StyleGAN2-ADA, thus I am not sure why this was not chosen as the baseline. Its one of the leading state-of-the-art methods and if the benefit of combining it with the proposed method is significant in all settings it will strengthen the paper further.
3. For the extremely limited setting for e.g., 0.1K FFHQ experiments, nearest neighbor test as done in [60] and latent interpolation visualization results should be shown as FID is not reliable in those settings. There might be overfitting and memorization of real images which is not evident from only quantitative evaluation.

---

> ### Author Response · Authors · 2022-08-02
> **Response to Reviewer SnfE**
>
> Thanks for the valuable comments. Individual concerns are addressed as follows. Note that, to make sure the revision is easy to track, we do not change the paper structure, and instead list all additional results in the supplementary material. We will rearrange the materials to make them fit in 9 pages in the next version.
>
> **Q1. Comparison to random channel selection.**
>
> A1: We copy some results in Tab. 1 below and add the result of the suggested baseline of random channel selection, where a subnet with half channels is randomly sampled from the original one at each training step. We confirm that such a random selection fails to alleviate the overfitting problem. Instead, our decreasing strategy brings more appealing performance.
>
> | FFHQ-$2K$                    | FID   |
> |--|--|
> | Baseline (full)                 | 78.82 |
> | Fixed half channels        | 63.36 |
> | Random half channels   | 78.22 |
> | Ours (full to half)            | 23.47 |
>
> **Q2. Why not use StyleGAN2-ADA as the direct baseline.**
>
> A2: Data augmentation is primarily proposed to improve GAN training with limited data. As reported by StyleGAN2-ADA [30], data augmentation may cause the augmentation leak problem, and also harm the performance given sufficient training data. The problem studied in this paper is more general, where we confirm that GAN training could benefit from a discriminator with *dynamic capacity* given not only limited training data but also sufficient training data. Hence, we choose StyleGAN2 as our direct baseline. But we do have shown the synergy of our approach and different techniques that improve GAN training in Tab. 5a. Thus our method which follows a different direction (varying model capacity) is diagonal to the data augmentation methods, which can be used together to further improve GAN training.
>
> **Q3. Quantitative evaluation on $0.1K$ FFHQ.**
>
> A3: We perform the nearest neighbor test *in Fig. S6 of the revised supplementary material (Page 5)*, and the latent interpolation visualization  *in Fig. S5 of the revised supplementary material (Page 5)*. The results show that the generator supervised by our dynamic discriminator does not memorize the dataset.
>
> **Q4. Training time and convergence curves.**
>
> A4: Under our default setting (*i.e.*, using half channels at the start of increasing strategy or at the end of decreasing strategy), it requires 15.97 hours when using 8 Nvidia A100 GPUs for training, while the baseline requires 18.75 hours. Our approach is around **15% faster**.
>
> We also include the time-varying FID curves *in Fig. S4 of the revised supplementary material (Page 5)*. We observe from the curves that, at the very start of training, our approach shows limited strength since not all the parameters get trained due to the subnet sampling. But after a short period of warming up, our approach outperforms the baseline consistently.
>
> **Q5. Difference of MetFaces fine-tuning experiments.**
>
> A5: The experiment on MetFaces in the original StyleGAN2-ADA paper [30] is conducted under the resolution of $1024 \times 1024$. In this work, our experiments are conducted under the resolution of $256 \times 256$. We clarify this experimental setting *in the revised version (Line 318)*.
>
> **Q6. The schedule of increasing alpha or decreasing beta is not mentioned in the paper. Is it linear and does having different schedules affect training? Also, decreasing channels schedule is only applied to higher level layers. What are the specific layer numbers and how does this choice affect the results?**
>
> A6: Yes, we linearly adjust $\alpha$ and $\beta$ (as in Line 144 and Line 157). We also conducted a new experiment to study the effect of different schedules. As shown *in Tab. S1a of the revised supplementary material (Page 6)*, the “cosine” schedule presents a similar result as the “linear” schedule, both of which surpass the baseline with a clear margin.
>
> In addition, *Tab. S1b of the revised supplementary material (Page 6)* investigates how many low-level layers are supposed to be excluded from decreasing channels. We can see that excluding the first 9 layers leads to the best result, which we choose as the default for all experiments that use the decreasing strategy.
>
> **Q7. Meaning of “we extract the numbers of adaptive dropout from ADA”.**
>
> A7: In Line 262, what we want to say is that the result is transcribed from the original StyleGAN2-ADA paper [30]. It is correct that StyleGAN2-ADA does not involve dropout in the discriminator. Instead, it uses the “adaptive dropout” variant as a baseline. We rephrase this sentence *in the revised version (Line 260)*.

---

### Official Review · Reviewer_XDah · 2022-07-11

**Rating:** 6
**Confidence:** 4
**Soundness:** 3 good
**Presentation:** 3 good
**Contribution:** 3 good

**Summary:**

In a standard GAN, a fixed-capacity discriminator is used during training. In contrast, this paper proposes DynamicD, which changes the network capacity of the discriminator dynamically during training by varying the network width. In particular, the authors introduce two strategies. The first strategy is to increase the capacity progressively. This strategy is proper when given sufficient data. The second strategy is to decrease the capacity gradually. This strategy is useful when given limited data. The effectiveness of the proposed method was demonstrated in various settings, including 2D and 3D-aware image synthesis. Furthermore, the compatibility with other techniques, such as data augmentation, regularizers, and pre-training, was shown.

**Questions:**

- What is the theoretical background of the proposed method?
- Does the proposed model outperform the model that uses a standard dropout where zero is inserted and the dropout rate increases gradually?
- Could you provide the time-varying performance analysis? I would like to know whether the proposed method outperforms the baselines constantly through training.
- Is it possible to provide the results for the blanks in Table 2?

**Limitations:**

In the paper, the exploration of the influence of different factors, such as network depth, applications to other architectures, such as a transformer-based discriminator, and development of the method for controlling the network capacity automatically, are described as future work.

In addition to the above, broader impacts (e.g., the risk of making it easy to create fake content, such as deepfake) are addressed in the supplementary material.

**Strengths And Weaknesses:**

**Strengths**
- The proposed ideas (i.e., increasing the capacity progressively and decreasing the capacity gradually) are simple. It will be easy to implement them.
- The effectiveness of the proposed method was demonstrated in various settings, including 2D and 3D-aware image synthesis. In the experiments, the proposed model was compared with multiple state-of-the-art baselines.
- The compatibility with other techniques, such as data augmentation, regularizers, and pre-training, was shown. The proposed method has high versatility.

**Weaknesses**
- The effectiveness of the proposed method was mainly demonstrated through an empirical analysis, and theoretical analysis was not sufficiently conducted. This paper would become more appealing if a theoretical analysis could be performed.
- The explanation about the implementation of a decreasing strategy is a bit rough. It is unclear what is the difference between the proposed method and the dropout where the dropout rate increases gradually. In Table 2, the results using a variant of dropout (adaptive dropout [30]) was presented. However, in the method, multiplicative Gaussian dropout was used. It would be more important to compare the proposed model with the model that uses a standard dropout where zero is inserted and the dropout rate increases gradually.
- The primary focus of this paper is to improve the training dynamics. Therefore, it would be better to analyze the time-varying performance in more detail. However, such an analysis is only conducted for the baselines in Figure 1. It would be better to perform the same analysis for the proposed method.
- There are many blanks in Table 2. This makes it difficult to judge whether the proposed model constantly outperforms or underperforms the baselines.
- Although the authors have already admitted, the model capacity must be adjusted manually. This is not an appealing property.

---

> ### Author Response · Authors · 2022-08-02
> **Response to Reviewer XDah**
>
> Thanks for the valuable comments. Individual concerns are addressed as follows. Note that, to make sure the revision is easy to track, we do not change the paper structure, and instead list all additional results in the supplementary material. We will rearrange the materials to make them fit in 9 pages in the next version.
>
> **Q1. This paper would become more appealing if a theoretical analysis could be performed.**
>
> A1: As you have already pointed out, our strategy is simple (*e.g.*, same hyper-parameters for all architectures and datasets) and effective (*e.g.*, improve the performances under both the sufficient data and the limited data settings). Providing a theoretical analysis for such a complex optimization problem is indeed very meaningful but posed to be challenging. It remains as a long-standing question in the neural network literature. We discuss this as a limitation of this work *in the revised version (Line 334)*. Still, we believe that our approach, together with its extensive empirical results, could inspire more theoretical studies in this direction.
>
> **Q2. Difference between our decreasing strategy and dropout.**
>
> A2: First, recall that our focus is to align the model capacity of the discriminator with the evolving difficulty of the bi-classification task in GAN training. “Decreasing strategy” is one scheme of our approach to deal with the insufficient data setting. Second, our strategy differs from the standard dropout in three main aspects: (1) We focus on the model capacity, and hence sample subnets from the discriminator backbone, forming a “weight-level” dropout. Differently, the standard dropout replaces features with zeros, forming a “feature-level” dropout. (2) Our “weight-level” dropout is shared by all instances, while the “feature-level” dropout is more like a per-instance regularizer. (3) In our strategy, only a sub-collection of parameters (*i.e.*, those of the sampled subnet) would be updated in each training iteration. We add this discussion *in the revised version (Line 157)* to clarify this difference.
>
> As requested, the table below shows the comparison between our approach against the standard dropout with a gradually increasing dropout rate, where we can see a clear performance gain.
>
> | FFHQ-$2K$                       | FID   |
> |--|--|
> | Standard dropout with increasing ratio  | 69.38 |
> | Ours                               | 23.47 |
>
>
> **Q3. Analyze the time-varying performance in more detail.**
>
> A3. We plot the requested time-varying FID curves under both the sufficient and the limited data regimes *in Fig. S4 of the revised supplementary material (Page 5)*. We observe from the curves that, at the very start of training, our approach shows limited strength since not all the parameters get trained due to the subnet sampling. But after a short period of warming up, our approach outperforms the baseline consistently.
>
> **Q4. Missing results in Table 2?**
>
> A4. Existing numbers are mostly extracted directly from the original papers.  In the revised version we try our best to provide some new results obtained by our implementation, and **highlight** them in Tab. 2.
>
> **Q5. Although the authors have already admitted, the model capacity must be adjusted manually. This is not an appealing property.**
>
> A5. For each strategy, our approach relies on a hyper-parameter for the dynamic adjustment of the model capacity. Choosing the most optimal value could be time-consuming, but we have shown that the training could already benefit from a somewhat adequate value. Recall that *all experiments in the submission share the same hyper-parameters*. We also provide the ablation studies on the hyper-parameters in *Tab. S1 in the revised supplementary material (Page 6)*. We leave it as future work to find an automatic mechanism for the adjusting hyper-parameter.

---

### Author Response · Authors · 2022-08-06
**Any more comments or concerns?**

Dear reviewers and AC

Thanks a lot for your effort in reviewing this submission! We have tried our best to address the mentioned concerns/problems in the rebuttal. Feel free to let us know if there is anything unclear or so. We are happy to clarify them.

Best,
Authors

---

### Meta-Review · Area_Chair_cuTf · 2022-08-26

**Recommendation:** Accept
**Confidence:** Less certain

**Metareview:**

This method uses a dynamic-capacity discriminator to improve GAN training, improving perforation eg in limited data settings. The method interested the reviewers, though there were common concerns about how the work was presented, both in what it is doing precisely as well as how it presents itself relative to prior works. This concern I share as well reading the reviews and glancing at the paper, yet this seems to have satisfied the reviewers, so I will go with their consensus though lower my confidence:

I therefore recommend (with uncertainty) that this paper is accepted to NeurIPS.

Reviewer  h4W2 participated the most in the discussion, but unfortunately discussion was overall light.

**Award:**

No

---

### Decision · Program_Chairs · 2022-09-14

Accept